

# Chlorophyll-*a* growth rates and related environmental variables in global temperate and cold-temperate lakes

5  Hannah Adams[1], Jane Ye[1], Bhaleka Persaud[1], Stephanie Slowinski[1], Homa Kheyrollah Pour[2], Philippe Van Cappellen[1]

[1] Ecohydrology Research Group, Department of Earth and Environmental Sciences and Water Institute, University of Waterloo, Waterloo, ON, Canada

[2] ReSEC Research Group, Department of Geography and Environmental Studies, Wilfrid Laurier University,
10  Waterloo, ON, Canada

*Correspondence to:* Hannah Adams (hadams21@mun.ca)

15  **Keywords:** lake primary productivity, chlorophyll-*a*, growth window, growth rate, bottom-up controls, trophic state, ice phenology, solar irradiance





**Abstract**

Lakes are key ecosystems within the global biogeosphere. However, the bottom-up controls on the biological productivity of lakes, including surface temperature, ice phenology, nutrient loads and mixing regime, are increasingly altered by climate warming and land-use changes. To better understand the environmental drivers of lake productivity, we assembled a dataset on chlorophyll-*a* concentrations, as well as associated water quality parameters and surface solar irradiance, for temperate and cold-temperate lakes experiencing seasonal ice cover. We developed a method to identify periods of rapid algal growth from *in situ* chlorophyll-*a* time series data and applied it to measurements performed between 1964 and 2019 across 357 lakes, predominantly located north of 40°. Long-term trends show that the algal growth windows have been occurring earlier in the year, thus potentially extending the growing season and increasing the annual productivity of northern lakes. The dataset is also used to analyze the relationship between chlorophyll-*a* growth rates and solar irradiance. Lakes of higher trophic status exhibit a higher sensitivity to solar radiation, especially at moderate irradiance values during spring. The lower sensitivity of chlorophyll-*a* growth rates to solar irradiance in oligotrophic lakes likely reflects the dominant role of nutrient limitation. Chlorophyll-*a* growth rates are significantly influenced by light availability in spring but not in summer and fall, consistent with a switch to top-down control of summer and fall algal communities. The growth window dataset can be used to analyze trends in lake productivity across the northern hemisphere or at smaller, regional scales. We present some general trends in the data and encourage other researchers to use the open dataset for their own research questions.

# 1 Introduction


Lakes play an important role in the biogeochemical cycling of many elements (Battin et al., 2008; Cole et al., 2007; O'Connell et al., 2020; Rousseaux and Gregg, 2013; Schindler, 1971). With over 100 million documented lakes on earth (Verpoorter et al., 2014), evidence indicates that the majority of global lakes are shallow with enough light and nutrients available to make

them highly productive ecosystems (Downing et al., 2006; Wetzel, 2001). Lakes therefore represent active sites for the storage, transport, and transformation of carbon, nutrients (e.g., nitrogen, phosphorus, silicon, iron), and contaminants (e.g., mercury) along the freshwater continuum (Lauerwald et al., 2019; Tranvik et al., 2009).

There are multiple bottom-up controls on lake primary productivity, including water

temperature, ice phenology, nutrient concentrations, circulation, mixing regime, and solar radiation (Lewis, 2011). Stressors such as climate change and nutrient pollution can significantly impact these controls, altering the ecosystem structure and biogeochemical functioning of lakes (Jeppesen et al., 2020; Markelov et al., 2019). Changes affecting northern lakes include warmer water temperatures, enhanced stratification and hypoxia, nutrient enrichment, light attenuation

by chromophoric organic matter, and increases in the relative abundance of toxic cyanobacteria in the phytoplankton community (Deng et al., 2018; Huisman and Hulot, 2005; Jeppesen et al., 2003; Creed et al., 2018). For example, Lake Superior has seen an increase in primary production during the last century, together with increasing surface water temperatures and longer seasonal stratification and ice-free periods (O'Beirne et al., 2017). Other lakes are

similarly experiencing increases in productivity. According to Lewis (2011), the current mean primary production of lakes is 260 g C $m^{-2}y^{-1}$, which is 162% higher than earlier estimations under historical baseline conditions.

Phytoplankton (i.e., algae) are the main primary producers in lakes and generally make up the foundation of lentic food webs (Carpenter et al., 2016). Periods of high lake productivity

coincide with a rapid increase in phytoplankton biomass. In extreme cases, algal blooms can reach hundreds to thousands of cells per milliliter (Henderson-Seller and Markland, 1987). These bloom events produce large quantities of decomposing organic matter that cause the expansion of hypoxic conditions within the lake (Watson et al., 2016). In harmful algal blooms, certain algal species also release hepatotoxic and neurotoxic compounds (Codd et al., 2005). Thus,

identifying trends in the timing and intensity of seasonal algal growth, and linking them to
changes in environmental stressors, can help predict the future of lake productivity and assess the
risk of undesirable algal blooms.

Because it is challenging to measure algal population growth directly, chlorophyll-*a* is often used
as a proxy for both the algae biomass and the associated primary production rate in lakes (Huot
et al., 2007). Although other proxies have been developed (Lyngsgaard et al., 2017), chlorophyll-
*a* is the most common metric to characterize trends in algal biomass within and across lakes,
especially in historical water quality records. Tett (1987) proposes a chlorophyll-*a* threshold of
$100\,\mu gL^{-1}$ to define "exceptional" blooms", Jonsson et al. (2009) use a threshold of $5\,\mu gL^{-1}$ to
identify a bloom, while Binding et al. (2021) flags an algal bloom when the chlorophyll-*a*
concentrations extracted from satellite observations exceed $10\,\mu gL^{-1}$. Such threshold values,
however, do not take into account the baseline (i.e., no-bloom) chlorophyll-*a* concentration
specific to a given lake, or the lake's trophic status (German et al., 2017). Furthermore, focusing
on harmful and nuisance algal blooms alone may mask the impact that a changing climate or
other stressors may have on a lake's overall biological productivity.

Annual fluctuations in lake chlorophyll-*a* concentration are an indicator of the natural seasonal
succession of algal species as a function of temperature, light, and nutrient availability
(Lyngsgaard et al., 2017). For instance, in a dimictic lake algal growth in the spring tends to be
controlled by these bottom-up controls, with light often being the primary limiting factor, while
later in the summer or fall algal biomass may be more influenced by zooplankton grazing (i.e., a
top-down control), while nutrient availability may overtake solar radiation as the limiting
resource for growth (Lewis et al., 2018; Lyngsgaard et al., 2017; Scofield et al., 2020).

A common approach for comparing chlorophyll-*a* trends across multiple lakes is to consider the
maximum or mean annual chlorophyll-*a* concentrations. For example, Ho et al. (2020) used the
Mann-Kendall trend test to analyze time series of annual maximum chlorophyll-*a* concentrations,
while Shuvo et al. (2021) used a random forest regression approach to assess the relative
importance of climatic versus non-climatic controls on mean chlorophyll-*a* concentrations.
However, these approaches do not specifically look at the periods of the year when algal biomass
is primarily determined by bottom-up controls and exhibits rapid growth.

Alternatively, the rate of change in chlorophyll-*a* concentration can be used to capture the timing
of rapid increase in algal biomass associated with periods of high lake productivity. In this study,
we refer to these periods as "growth windows". The weeks leading up to a growth window are
crucial to create the necessary environmental conditions that enable algal growth (Lewis et al.,
2018). Thus, to analyze trends in lake productivity one should consider environmental variables,
such as surface water temperature, solar radiation and nutrient concentrations, both during and
preceding the annual growth windows.

Although the rate of chlorophyll-*a* concentration growth has been used to detect algal blooms
within individual water bodies, for example in the San Roque reservoir (Germán et al., 2017), it
has rarely been used across large temporal (i.e., more than a few years) and spatial (i.e., regional
and up) scales. Here, we present a method for calculating seasonal chlorophyll-*a* growth rates
and then create a dataset of these rates derived from *in situ* chlorophyll-*a* concentrations obtained
in 357 lakes, most of which are at latitudes above 40° N. The entire dataset covers the period
from 1964 to 2019, and further contains data on coincident bottom-up environmental control
variables, including *in situ* surface solar radiation measurements. To illustrate the potential
applications of the dataset, we present some general trends of the chlorophyll-*a* rates and their
relationships with environmental variables. The dataset is made available as an open resource
that other researchers are encouraged to use in their own work.

## 2 Data and methods

All data processing, visualizations, and analyses were carried out with Python (ver. 3.7.6; Python
Software Foundation, 2021) using the pandas library (Reback et al., 2020), NumPy library
(Harris et al., 2020), and Dplython library (Riederer, 2015), while QGIS/PYQGIS was used for
all spatial data analyses (ver. 3.16; QGIS Development Team, 2021).

### 2.1 Data acquisition, compilation, and quality control

### 2.1.1 Lake data selection

*In situ* chlorophyll-*a* concentrations and other lake physico-chemical data were collected from
open source international, national, and regional databases. The data include surface water
temperature, Secchi depth and pH, as well as the concentrations of particulate organic carbon
(POC), total phosphorus (TP), soluble reactive phosphorus (SRP), total Kjeldahl nitrogen (TKN)
and dissolved organic carbon (DOC). We selected lakes from latitudes ≥ 40° N to reduce the

latitude-dependent variability in mixing and thermal regimes, both of which exert a strong
control on lake productivity (Kirillin et al., 2012). At mid-to-high latitudes most lakes are
dimictic with seasonal ice cover while low-latitude lakes are typically meromictic or polymictic
(Woolway and Merchant, 2019). High-elevation lakes at lower latitudes can experience similar
effects from the transition from winter to spring, even without ice cover (Deng et al., 2020). We
therefore included the extensively monitored Lake Kasumigaura in Japan and Lake Taihu in

China in our study, although they are located at latitudes lower than 40°N.

Chlorophyll-*a* measurements are collected at variable water depths by different lake monitoring
agencies and researchers. For consistency, we only included measurements taken at ≤ 3 m depth.
When the sampling depth was not provided, we assumed the sample was taken from within the
top 0.5-3 m of the lake, given that this is standard sampling protocol (Dorset Environmental

Science Centre, 2010; United States Environmental Protection Agency, 2012).

We omitted all variable values below the corresponding analytical detection limit. Data from
different sources were individually reformatted to yield consistent (standard) units and headings.
Reported values were averaged to yield daily values mean before being combined into a single
csv file. When multiple chlorophyll-*a* data types were available (as, for example, in the

Laurentian Great Lakes data series), we selected the uncorrected data because most reported lake
chlorophyll-*a* concentrations have not been corrected for phaeophytin pigments. If no
coordinates were provided, we assigned those of the lake centroid in QGIS or estimated based on
the location name. Fifteen lakes had no known location and were removed from the final dataset.
We further restricted ourselves to lakes that were sampled at least 8 times per year. This was

found to be the minimum number of sampling points required to detect the growth windows. The
location of all lake sampling locations in the growth window dataset are shown in Figure 1.

After the above selection and quality assessments, the final dataset used for calculating the
growth windows contained 52116 unique data points (62% of the original data) for 357 lakes, all
≥ 40°N (except Lake Kasumigaura and Lake Taihu), covering the period 1964-2019.

**2.1.2 Surface solar radiation data**

Open source *in situ* surface solar radiation (SSR) data for the period 1950-2020 were collected
from stations paired with the selected lakes. Each lake was paired with the closest SSR station
using the nearest neighbor function in QGIS, allowing for a maximum radius of three degrees



(Schwarz et al., 2018; Figure 1). In the dataset, the geodesic distance between each lake and its
paired SSR station is given, as well as the difference in elevation.

The SSR data temporal resolution varied from minutes to months. Hence, where needed, the SSR
data were resampled to yield monthly mean values. For the Experimental Lakes Area (ELA) in
Ontario, Canada, the data were converted from photosynthetically active radiation (PAR) to
SSR, where the PAR wavelength range (400-700 nm) was averaged to 550 nm.

### 2.1.3 Lake characteristics

For each lake, we calculated the trophic status index (TSI) based on the mean chlorophyll-*a*
concentration over the sampling period. This TSI value was used to assign the lake to the
corresponding trophic state category according to Carlson and Simpson (1996). The
HydroLAKES shapefile yielded the lake's surface area, mean depth, elevation, and volume
(Messager et al., 2016). The climate zone of the lake was extracted from the HydroATLAS
shapefile (Linke et al., 2019).

### 2.2 Detecting seasonal growth windows

Growth windows were defined based on the rate of change in chlorophyll-*a* concentration at each
lake sampling point throughout the year. To locate the start and end of a growth window, we
smoothed the annual chlorophyll-*a* time series using a Savitzky-Golay filter (SciPy.signal
savgol_filter) and flagged optima in the smoothed data (SciPy.signal find_peaks) using functions
from the open source SciPy ecosystem (Virtanen et al., 2020). The procedure is illustrated in
Figure 2.

For each year, the spring growth window began when the daily rate of increase surpassed the
threshold of 0.05 µgL$^{-1}$day$^{-1}$ for the first time. The 0.05 µgL$^{-1}$day$^{-1}$ rate was chosen because it
corresponds to the median rate at which a distinct switch to a "rapid growth" period in the
mesotrophic-hypereutrophic lakes in the dataset was observed. The growth window ended at the
first "peak" in chlorophyll-*a* concentration. If a threshold rate of 0.05 µgL$^{-1}$day$^{-1}$ was never
reached during a given year, the growth window began when the rate of change first became
positive. The summer (or fall) window was identified in the same way following the end of the
spring window. If there was only one peak in the smoothed data, only one growth window was
identified for that year. This year was then labelled as a "single growth window" year (i.e., only



one major algal growth window occurred within that year). Years with more than three chlorophyll-*a* peaks, or with no peaks, were not included in the growth window dataset.

Depending on data availability, the pre-growth window was defined as the one or two week period immediately preceding the growth window start date. For each pre-growth window, the mean surface water temperature, SSR, and TP concentration were calculated. These served as (simple) indicators of how favorable in-lake conditions were to initiate algal growth (Lyngsgaard et al., 2017). An example of a spring and summer growth window is shown in Figure 3. Note that

we use the label "summer" to indicate the second yearly growth window, although in many cases the summer growth window occurred after the fall equinox.

Once the growth window and pre-growth durations were determined, the mean values of the variables listed in Table 1 were calculated for both the growth window and the pre-growth window. This was done for each lake and for each year data were available. In the dataset, each

row represents a single growth window and includes the timing and duration, rate of increase of the chlorophyll-*a* concentration, and all other relevant lake variables, including SSR. Note that, along with the variables in Table 1, we included the total number of samples collected each year so the dataset can be filtered for sampling frequency. The reader is referred to the supplementary information included with the dataset for a more detailed explanatory table with additional

information on the organization carrying out the monitoring, physiological attributes of each lake, and years that data are available for a given sampling location.

## 3 Dataset: data distributions

### 3.1 Dataset characteristics

Most lakes in the dataset are located between 50 and 60° N as the majority of available open data

are from organizations within the United Kingdom, Sweden, Canada, and the United States. The years with available data in the dataset are unevenly distributed, however, with most detected growth windows falling in the period 2005-2019, likely due to a combination of increased lake monitoring efforts and a push in recent years towards greater accessibility of publicly funded data (Hallegraeff et al., 2021; Roche et al., 2020; Figure 4a).

The majority of growth windows recorded in the dataset fall in the eutrophic category (1.6% oligotrophic, 18.0% mesotrophic, 75.4% eutrophic, and 5.0% hypereutrophic). Single growth

windows dominate oligotrophic lakes where they make up 96% of all growth windows (Figure 4b). This may reflect the severe nutrient limitation in oligotrophic lakes, which prevents the occurrence of a second annual algal growth window (Rigosi et al., 2014). Oligotrophic lakes also tend to occur at the higher latitudes (Figure 4c) where lower water temperatures and solar radiation may further limit algal growth (Lewis, 2011).

The growth window durations range from 2 to 251 days, with a median of 71 days across all lakes (Figure 5a). Summer growth windows tend to be shorter than those of spring and single growth windows, with the latter exhibiting the most variable start and end dates (Figure 5b).

## 3.2 Environmental conditions during growth windows

Chlorophyll-*a* rates during the growth windows exhibit log-normal distributions (Figure 6a). The mean chlorophyll-*a* rate is lowest in the single growth window category and highest in the summer growth windows. Mean surface water temperature has a distinct bimodal spring-summer distribution (Figure 6b), which is expected for northern temperate and cold-temperate lakes where surface water temperature during the ice-free period follows the seasonal air temperature trend (Kirillin et al., 2012). For the single growth windows, temperature is evenly distributed across the annual range, which aligns with the large variability in the timing of single growth windows (Figure 5b). Total phosphorus concentrations are lowest during the spring growth windows, which likely reflects a greater control of P limitation on algal growth during spring compared to summer and fall (Kirillin et al., 2012; 6c). Secchi depth during the growth windows ranges from 0.01 to 14.6 m, with summer growth windows experiencing the lowest mean Secchi depth, as turbidity generally increases after the spring bloom (Figure 6d).

## 4 Dataset: trend analyses

The growth window delineation and the estimation of chlorophyll-*a* rates can in principle be applied to any lake for which time series chlorophyll-*a* concentration data are available. By creating a dataset comprising many lakes and covering multi-year time periods, it becomes possible to analyze global trends in lake productivity. Here, we provide a few illustrative examples of how the dataset can be interrogated, thereby setting the stage for its use by other researchers.





### 4.1 Chlorophyll-*a* rates: trophic status and latitude

When grouped by trophic status, mean and median chlorophyll-*a* growth rates show the expected increase from oligotrophic to hypereutrophic lakes (Figure 7a). The rates in the different trophic categories, however, cover very large and overlapping ranges. When grouped according to latitude, lakes between 40 and 50° N exhibit the widest range in chlorophyll-*a* rates (Figure 7b) that, in part, reflects the high proportion of lakes in this latitude range. The highest latitude lakes (60-70° N) tend to have the lowest chlorophyll-*a* rates, which is expected given the cooler temperatures and lower solar irradiance they experience (Lewis, 2011).

While differences in chlorophyll-*a* rates usually indicate comparable differences in algal biomass growth rates, it is important to note that the chlorophyll-*a* to biomass ratio varies within and among lakes. In particular, chlorophyll-*a* to biomass ratios are known to be sensitive to variations in solar irradiance and temperature (Behrenfeld et al., 2016). The summer ratio of chlorophyll-*a* to biomass (typically expressed as particulate organic carbon concentration) generally decreases with increasing latitude because the algae are adapted to the more variable daylight conditions, including longer summer photoperiods, at higher latitudes (Behrenfeld et al., 2016). By contrast, cooler temperatures at higher latitudes may result in higher chlorophyll-*a* to biomass ratios because of lower growth rates, at least when the algae are nutrient-replete (Behrenfeld et al., 2016).

### 4.2 Chlorophyll-*a* rates: temperature and climate warming

The start and end dates of the spring, single and summer growth windows show temporal trends towards occurrence earlier in the year (Figure 8a). The trends are most pronounced for the spring windows, which likely reflects a greater sensitivity of springtime algal activity to climate warming. The latter causes earlier ice break-up and produces earlier surface water temperature conditions favorable for algal growth (Markelov et al., 2019). This hypothesis is consistent with the correlations between the chlorophyll-*a* rates and water temperature (Figure 8b).

The start and end dates of the spring growth windows show a positive correlation with increasing temperature (Figure 8b). By contrast, little or even negative correlations are seen for the summer growth windows. Thus, all other conditions unchanged, a warmer climate would see earlier spring blooms, but little temporal shifts for the summer growth windows and, possibly, even a slight delay. For the spring and single growth windows, the duration of the window shows a





maximum around 10° C. Therefore, moderate temperatures close to 10° C should, on average, produce the longest lasting algal growth events. No dinstinct trend is seen for summer growth windows, presumably because they occur when water temperatures are already above 10° C.

### 4.3 Chlorophyll-*a* rates: solar irradiance

Solar radiation is essential for phytoplankton growth (Inomura et al., 2020). For example, at the
single lake scale, Tian et al. (2017) showed that SSR is a major predictor of growing-season chlorophyll-*a* concentrations in the Western Basin of Lake Erie. A paleolimnological study of Lake Tanganyika also provided evidence for a positive correlation between multi-centennial oscillations of SSR and diatom productivity dating back to ~1000 CE (McGlue et al., 2020). Nonetheless, the relationship between algal growth and SSR has yet to be compared across a
large set of lakes.

Solar radiation is used directly by photosynthetic organisms for carbon fixation (Melkozernov and Blankenship, 2007). In addition, SSR exerts a strong control on lake surface water temperature (Jakkila et al., 2009) and the timing of ice breakup in seasonally ice-covered lakes (Kirillin et al., 2012b), both of which influence lake primary productivity. While the global
distribution of mean annual SSR is primarily a function of latitude (Kirillin et al., 2012b), atmospheric controls (e.g., cloud cover) cause regional variability, as well as variability over time (Alpert and Kishcha, 2008; Cutforth and Judiesch, 2007; Wild, 2009). It is important to note that SSR is not related directly to global warming (Kirillin et al., 2012b), nor is it controlled by the cycles in the sun's energy output (Wild, 2009).

To determine to what extent SSR explains variations in chlorophyll-*a* growth rates, we removed the effect of temperature by normalizing the rates using the temperature dependency function (which we refer to as "*ftemp*") proposed by Rosso et al. (1995). This function describes the non-linear temperature dependence of cellular metabolic activity and requires that a minimum, maximum, and optimum growing temperature be assigned. Dividing the *in situ* chlorophyll-*a*
rate during the growth window by the corresponding *ftemp* value corrects for the effect of differences in temperature between growth windows.

The temperature-corrected chlorophyll-*a* growth rates indicate that the relationship between SSR and algal growth is a function of the trophic status (i.e., nutrient availability), as seen in Figure 9. Lakes of higher trophic status are more sensitive to SSR than lakes of lower trophic status. For





eutrophic lakes, the effect of SSR on the temperature-corrected chlorophyll-*a* rates is most
pronounced in the low to moderate SSR range typical of the spring season (Figure 9a). The same
effect is not seen when considering the rates without temperature correction (Figure 9b). Thus,
the increasing SSR during spring is counterbalanced by cooler temperatures compared to the
later summer growth window. Note that the summer chlorophyll-*a* growth rates show little
influence from SSR, whether corrected or not for temperature, supporting the theory of a greater
top-down control on algal growth during the summer versus the spring as proposed, among
others, by Lyngsgaard et al. (2017).

The chlorophyll-*a* growth rate data near or above 200 Wm$^{-2}$ remain low, with no clear
dependence on SSR.  This is likely indicative of a photoacclimation response of the algae, where
they produce less chlorophyll-*a* in proportion to their total biomass so they can allocate more
resources to growth when nutrients – not light – are limiting growth (Lewis et al., 2018; Inomura
et al., 2020). Furthermore, when light intensity during the summer months reaches damaging
levels, algae may start producing additional photosynthetic pigments to protect their chlorophyll
(so-called sunscreen pigments). However, nutrient availability may limit the amount of pigments
that can be synthesized, impeding the photoacclimation response (Lewis et al., 2018). This
nutrient limitation of the photoacclimation response would explain the differences in the
temperature corrected growth rate's sensitivity to SSR as a function of trophic status (Figure 9a).
Lakes of higher trophic status (i.e., less nutrient limitation) show a larger response to changes in
SSR, presumably because they have sufficient nutrients to produce additional chlorophyll-*a* in
response to an increase in SSR.

## 5 Key findings

The following points summarize the general trends that emerged from our analysis of the dataset.

1. Higher water temperatures and reduced ice-cover cause algal growth windows to start earlier
   in the year, extending the growing season and potentially increasing annual net primary
   productivity of northern lakes under ongoing and future climate warming.
2. Chlorophyll-*a* growth rates increase with nutrient availability while they decrease at higher
   latitudes due to cooler temperatures and lower SSR.
3. Oligotrophic lakes tend to have the highest proportion of single annual growth windows,
   likely reflecting the dominant role of nutrient limitation.



4.  Temperature-corrected chlorophyll-*a* growth rates suggest a relationship with SSR that depends on the trophic state of lakes:

        a.  compared to mesotrophic and oligotrophic lakes, eutrophic lakes exhibit a higher sensitivity to SSR, especially in the low to moderate irradiance levels experienced during spring;

340        b.  at the upper end of SSR, chlorophyll-*a* growth rates remain low and independent of SSR, which may reflect a photoacclimation response of algae.

    5.  The low SSR sensitivity of chlorophyll-*a* growth rates during summer and fall suggests a stronger top-down control on algal growth compared to the earlier spring growth windows.

    6.  In summary, light limitation is an important control on chlorophyll-*a* growth rates during

345        spring, whereas lower nutrient availability and increased grazing from zooplankton tend to be more significant during summer.

## 6 Conclusions

We present a novel way to delineate periods of rapid algal growth, or growth windows, in lakes based on time series chlorophyll-*a* measurements. We apply this approach to derive the

chlorophyll-*a* growth rates occurring during the growth windows of 357 lakes from cold and cold-temperate regions in the northern hemisphere, using data collected between 1964 and 2019. The derived growth rates are assembled in an open-source dataset, together with additional information on the lakes including data on water quality, trophic state, and solar radiation. Note that the dataset can be paired with databases such as the HydroLAKES, HydroATLAS and

GLCP databases to access additional lake and/or watershed attributes. Our dataset is designed to support comparative analyses of the controls on algal productivity within and between lakes. We present several examples of such analyses. We hope these will encourage others to use the dataset in their own research and to further expand the dataset's geographical reach and information content.

**Code and data availability**

All code is available in the project GitHub repository (https://github.com/hfadams/growth_window) and in Zenodo (https://doi.org/10.5281/zenodo.5171442). The growth window dataset and supplementary data




files are available in the Federated Research Data Repository at

https://doi.org/10.20383/102.0488 (Adams et al., 2021).

## Author contributions

All authors took part in development of the study. SS, BP, and HKP conceptualized the study,

while HA and JY developed methods and carried out data collection and data post-processing.

HA wrote the original manuscript with contributions from JY, BP, SS, HKP, and PVC. All

authors reviewed and edited the final paper.

## Competing interests

The authors declare that they have no conflict of interest.

## Acknowledgments

This work is funded by the Canada First Research Excellence Fund's Global Water Futures

Programme. We would also like to thank all institutions listed in the supplementary information

for making their data open source and free to support this research.





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





*Table 1: Summary of variables in the derived growth window dataset.*

| Variable | Units | Description | Comments |
|---|---|---|---|
| **Season** | NA | Time of year when the growth window is detected: Spring, Summer, or Single growth window | A single growth window occurs when there is no decrease in chlorophyll-*a* concentration between spring and summer |
| **Chlorophyll-*a* growth rate** | µgL⁻¹day⁻¹ | Rate of increase in chlorophyll-*a* concentration between the start and end of the growth window | Chlorophyll-*a* rate of change between sampling times are used to define the growth window period |
| **Specific chlorophyll-*a* rate** | day⁻¹ | Chlorophyll-*a* growth rate divided by initial concentration | Can be compared across lakes |
| **Temperature-corrected specific chlorophyll-*a* rate** | day⁻¹ | Temperature correction function provided in supplementary data | Initial parameters: min temp=0°C, max temp=40°C, optimal temp=25°C |
| **POC growth rate** | mg L⁻¹day⁻¹ | Rate of increase or decrease between consecutive sampling times | Representative metric for the rate of change in total algal biomass |
| **Chlorophyll-*a* growth rate : POC growth rate** | mg chlorophyll-a : mg POC | Ratio of the chlorophyll-a and POC rates of change | Can be used to see how the chlorophyll-*a* rate of production changes in proportion to total algal biomass |
| **Mean surface water temperature** | °C | Mean value across the growth window and the 14-day pre-growth window | |
| **Surface solar radiation** | Wm⁻² | Mean value across the growth window and the 14-day pre-growth window | |
| **TP** | mg L⁻¹ | | (co-)limiting macronutrients |
| **SRP** | mg L⁻¹ | | |
| **TKN** | mg L⁻¹ | Growth window mean values | |
| **Secchi depth** | m | | Proxy for turbidity |
| **pH** | pH units | | |
| **Trophic Status Index (TSI)** | Range: from 0-100 | Calculated from the mean chlorophyll-*a* concentration across all years the lake was sampled | Used to assign trophic status |
| **Trophic status** | NA | Trophic status class assigned based on TSI: Oligotrophic, Mesotrophic, Eutrophic, or Hypereutrophic | TSI thresholds are those of the North American Lake Management Society |



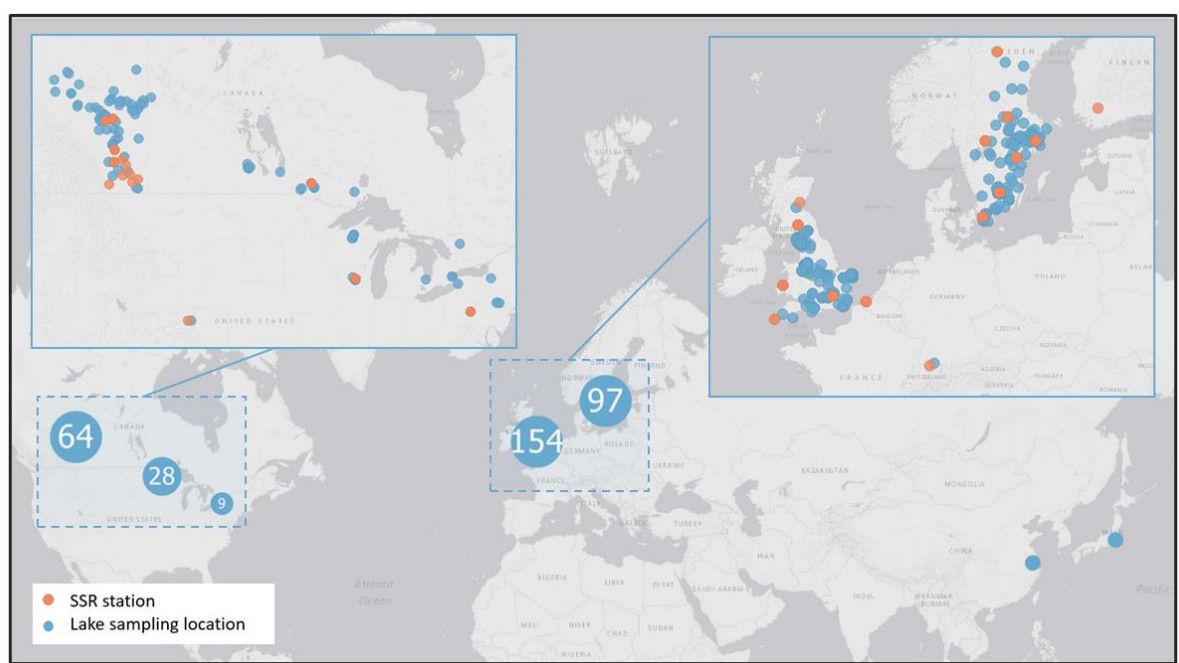

**680**   *Figure 1: Distribution of the 357 lake sampling locations in the growth window dataset. Sampling points are clustered by proximity, where marker size and value indicate the number of unique locations represented by each point. Enlarged sections show each lake sampling location and along with the location of the 322 paired SSR stations. Base map credit: ESRI, 2011.*

**685**

**690**



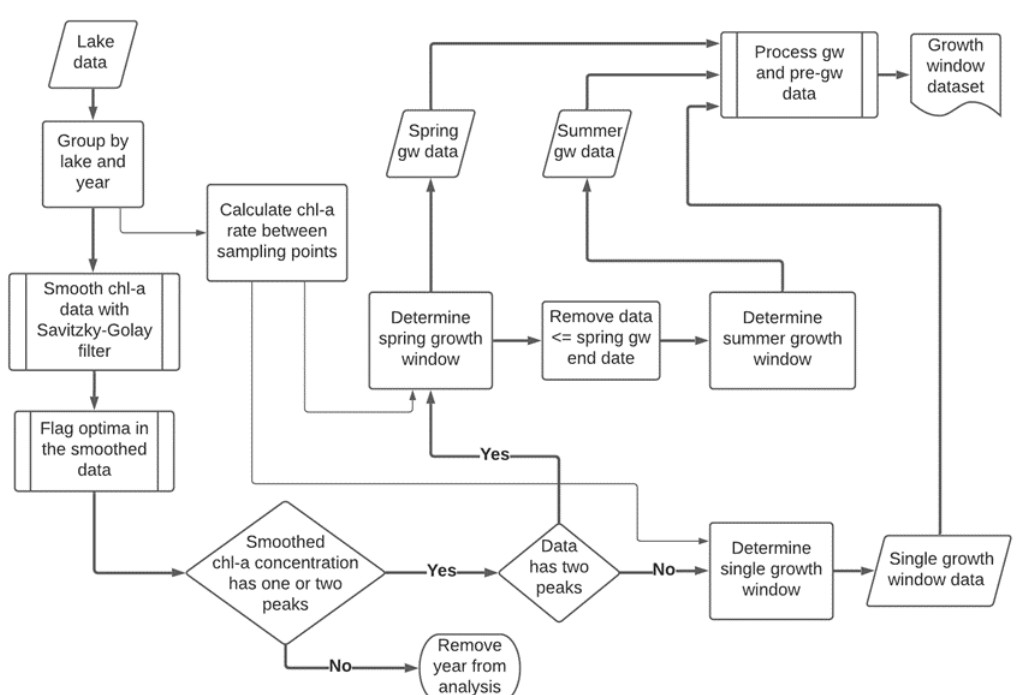

*Figure 2: Workflow for detecting and processing growth window data. For each lake sampling point, chlorophyll-a (Chl-a) data are smoothed with a Savitzky-Golay filter and then growth windows are detected based on peaks in the chlorophyll-a concentration. Growth windows are flagged as spring, summer, or single growth windows.*





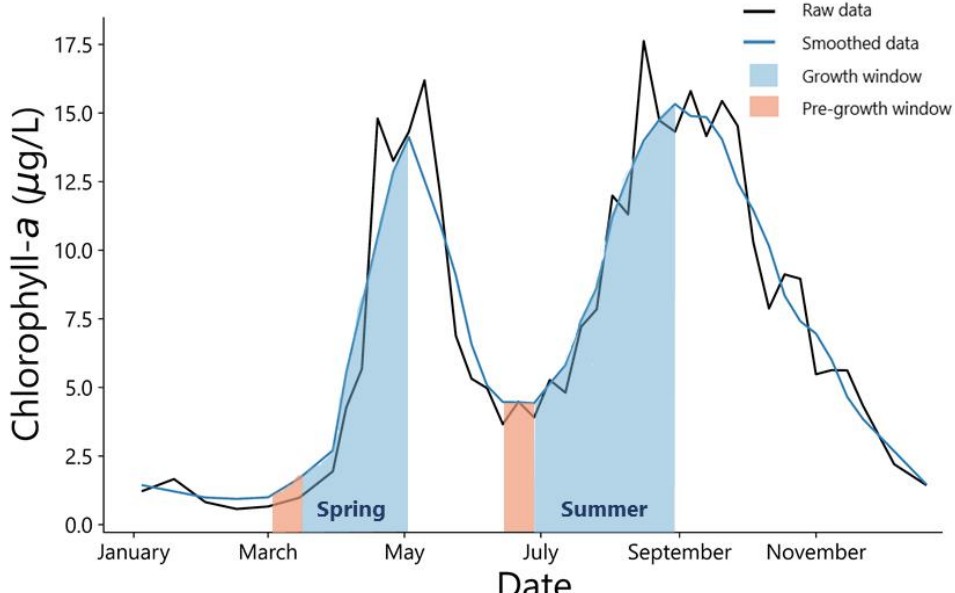

*Figure 3. Example of spring and summer growth windows in Lake Windermere's north basin in 1988. Peaks in the smoothed data indicate the end of the growth window, and the window begins when the rate of increase in chlorophyll-a concentration surpasses a threshold of 0.05 $\mu gL^{-1}day^{-1}$ (median rate for the distinct switch to a "rapid growth" period in mesotrophic-hypereutrophic lakes) for the first time. The growth window and pre-growth window (two weeks leading up to the growth window) are shown in blue and orange shading respectively.*


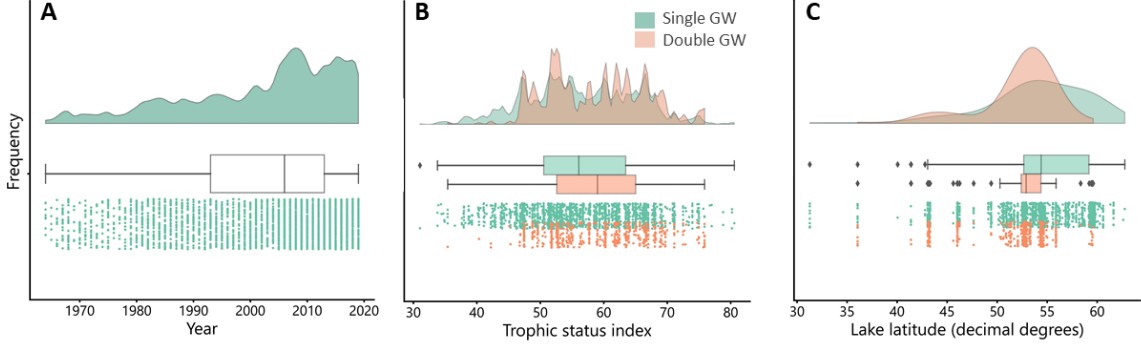

*Figure 4.  Distributions of (a) year of occurrence, (b) lake trophic status index, and (c) lake latitude for each growth window in the dataset. Data are grouped by "double GW" or "single GW" year. The data is skewed toward more recent years and higher latitudes. Lakes in the oligotrophic category (TSI < 40) have the highest proportion of single growth windows.*




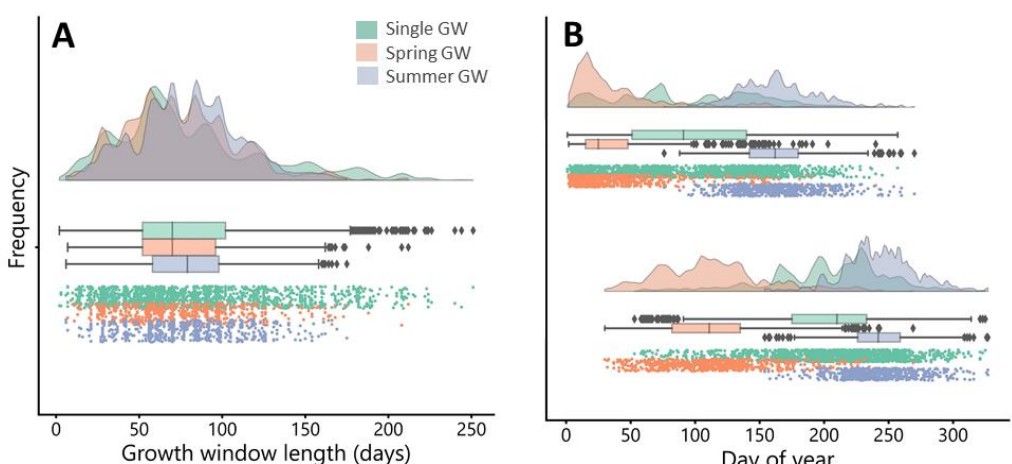

*Figure 5. Distributions of (a) duration and (b) timing of the growth windows, grouped by growth window type. Single growth windows have both the longest range in length and the most even distribution of start and end dates.*




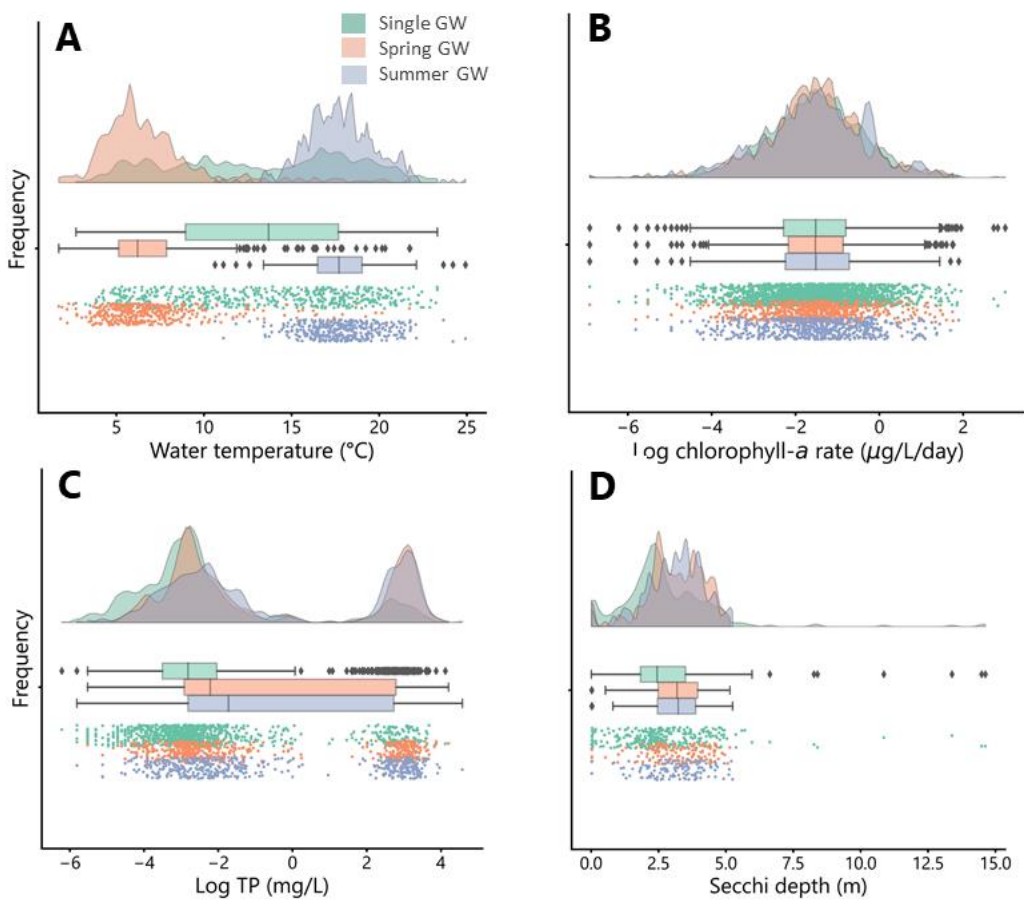

*Figure 6. Distributions of selected water quality variables during the growth window period: (a) log chlorophyll-a rate, (b) mean water temperature, (c) log mean total phosphorus (TP), and (d) mean Secchi depth.*





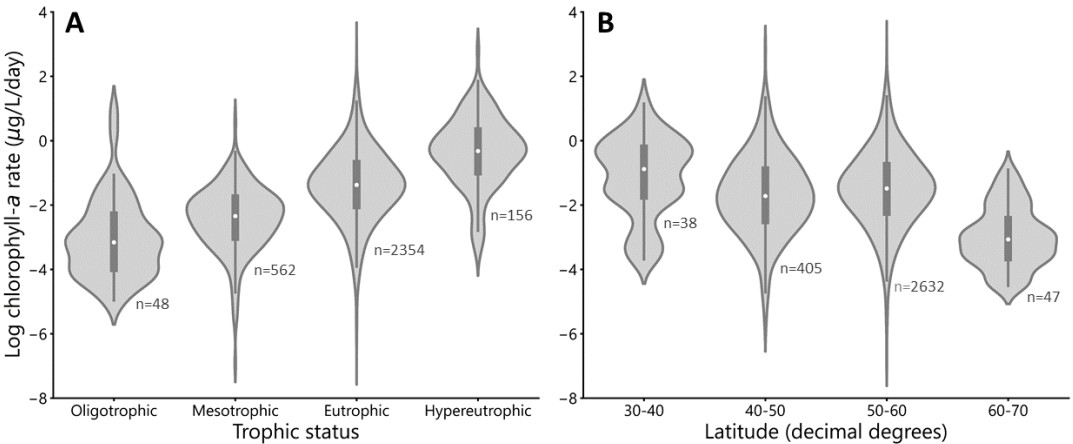

*Figure 7. Chlorophyll-a growth rate trends in the dataset: grouped by (a) trophic status and (b) latitude. Lakes of a higher trophic status have higher mean chlorophyll-a growth rates and lakes at higher latitudes have lower chlorophyll-a growth rate during the growth windows. The number of lakes represented by each violin is shown in text on the panels.*





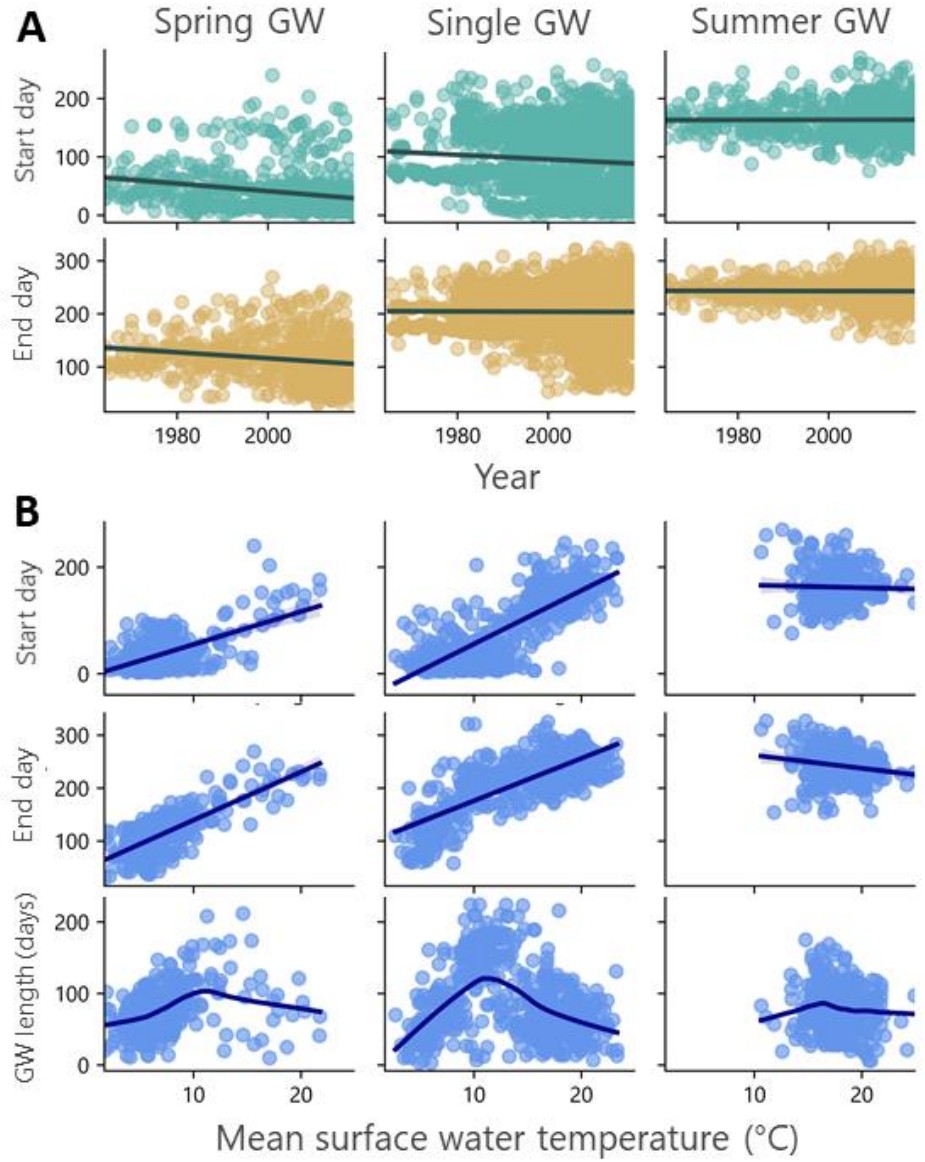

*Figure 8. (a) time series of the start and end dates for the spring, summer, and single growth windows for all the lakes in the dataset; all growth window categories trend toward earlier start and end dates, especially in the spring. (b) Start and end dates of the growth windows as a function of temperature (regression line in dark blue) suggest a positive relationship between growth window timing and surface water temperature in the spring and a negative relationship in the summer. Growth window length (dark blue trendline shows locally weighted scatterplot smoothing) shows that longer growth windows occur at moderate surface water temperatures that aren't seen in the summer months.*




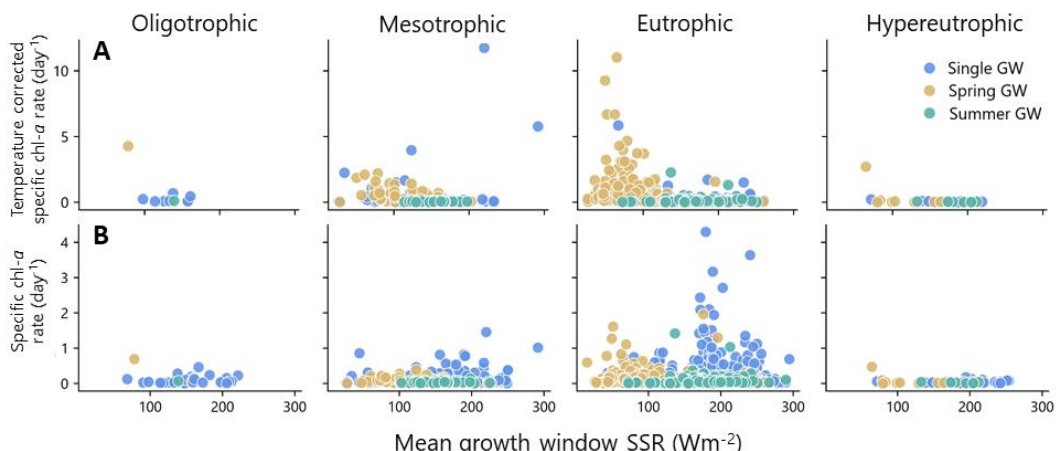

*Figure 9. Comparison of trends in the relationship between mean growth window SSR with (a) temperature corrected chlorophyll-a rate and (b) specific chlorophyll-a rate without temperature correction. Data are grouped by trophic status, and hue indicates growth window type. Lakes of a higher trophic status show an increased sensitivity to solar radiation, especially during the spring (panel A) while summer growth windows do not show sensitivity to solar radiation or water temperature, suggesting top-down control from zooplankton grazing. Low chlorophyll growth rates at SSR near or greater than 200 Wm⁻²*

*indicate a photoacclimation response in the algae.*