# Peer review of "Rates and timing of chlorophyll-*a* increases and related environmental variables in global temperate and cold-temperate lakes"

_Earth System Science Data, 2021_

## Author Comment (AC1)

**Comment on essd-2021-329**
Response to Referee #1
* * *
Referee comment on "Chlorophyll-*a* growth rates and related environmental variables in global temperate and cold-temperate lakes" by Hannah Adams et al., Earth Syst. Sci. Data Discuss., https://doi.org/10.5194/essd-2021-329-RC1, 2022
* * *
Review of Adams et al., Chlorophyll-a growth rates and related environmental variables in global temperate and cold-temperate lakes. ESSD-2021-0329

**General:**

This well-written data paper collates and integrates several commonly used open access water quality data sets (time series), including those from Fisheries and Oceans Canada (now IISD) Experimental Lakes Area, the US EPA, and the US Northern Lakes LTER Program, as well as substantial datasets from Alberta (Canada), England (UK), and Sweden. The authors summarize some, but not all, of the main features of the data, and provide some illustrative relationships among in situ and external environmental parameters designed to stimulate further analysis. One key feature of the data is the calculation of daily Chl a 'growth rates" (really, rate of change) that the authors use as a surrogate for phytoplankton population change. By identifying periods of growth > 0.5 ug Chl a/L/day, the authors examine how patterns of growth vary as a function of lake trophic status, irradiance, and limnological parameters both before and during the windows of growth. This analysis is both the strength and weakness of the paper, as the protocol does not appear to be used a prior peer-reviewed methods or application paper and is insufficiently validated in the current submission to allow its reliability or utility to be determined. Basically, the paper appears half-way between a simple data paper and a methods paper, and so is not really a good fit to either category at the moment. Key comments are outlined below, arranged according to the line number of the paper.

*We thank the reviewer and appreciate their suggestions and comments for improving our manuscript; all their suggestions have been considered in the revision of the manuscript. Below, we provide the answers to the comments and questions raised by the reviewer and all the modifications that have been incorporated in the revised version of the article.*

**Details**:
Terminology, l 49 and elsewhere. I think the authors would be better served to use less jargon and be more precise in their description of regulatory processes. Phrases such as "Bottom-up" and 'top down" appear derived from older food-web literature, and are used in a non-standard manner in the current paper. Here, 'bottom-up' appear to mean physical and chemical controls (physico-chemical), whereas 'top-down' seem to refer to grazing by zooplankton. I suggest they use these more precise description, as the current terms lack context when only one trophic level (phytoplankton) is actually described (as Chl a).

*We agree with the reviewer. We removed unnecessary jargon and used the descriptions suggested by the reviewer (e.g., zooplankton grazing rather than top-down control). For example, on lines 80-83, we state "Intra-annual fluctuations in lake chlorophyll-a concentration result from the interactions of multiple variables and processes including grazing by zooplankton, competition between algal species with different growth strategies and chlorophyll-a contents, and changes in temperature, light, and nutrient availability."*

Similarly, this appears to be a paper about the rate of change of Chl a standing stock/crop, not production (an amount) or productivity (a rate) per se. Its ok to note that changes in Chl are used to infer these other processes, but try and focus this paper on Chl to avoid over-extrapolation of the analyses (see below). This gets to be an issue when discussing the 'two phases' of change in Chl, separated by what is commonly known as
the clearwater phase (CWP). The latter occurs in productive (but rarely oligotrophic) lakes and arises when thermal stratification coincides with development of populations of large- bodies zooplankton (often *Daphnia*) to greatly reduce spring biomass maxima due to sedimentation (mainly diatoms) and herbivory (usually diatoms, flagellates). This decline in Chl does not necessarily correspond to declines in the rate of production (productivity), as available nutrients are usually elevated (because of recycling of biomass via
homeostatic herbivores). Jim Elser and Bob Sterner have done a lot of work on this (see "ecological stoichiometry"). The main point is that rates of change in Chl do no necessarily correspond to changes in primary productivity, particularly through a typical annual phenology.

*We fully agree that the rate of change of the chlorophyll-a concentration is not solely related to algal productivity. To avoid any ambiguity, we modified the terminology used in the manuscript. The term "growth" is no longer used, and the following clear definitions are now given in the text. Specifically, we have renamed:*

- *"Growth rate" as "rate of chlorophyll-a increase" (RCI), defined on line 115-116*
- *"Growth window" as "period of chlorophyll-a increase" (PCI), defined on line 107*
- *"Specific growth rate" as "normalized rate of change in chlorophyll-a concentration" (NRCC), defined on lines 182-183*

*Renaming the "specific growth rate" to "normalized rate of change in chlorophyll-a concentration" (NRCC) avoids the assumption that the rate of increase in chlorophyll-a concentration must be strictly proportional to the increase in algal biomass concentration.*

*We now avoid any unwarranted interpretation and provide cautionary statements where needed. We have reworded our explanation about what the calculated rate of chlorophyll-a increase (RCI) represents and what the limitations of this metric are in lines 263-276. We have also highlighted in the text that the normalized rate, NRCC, is a relative rate which facilitates the comparison between lakes of different trophic status, whereas the RCI will vary systematically between lakes with different standing stocks of chlorophyll-a (lines 263-276). We have calculated the NRCC to use as the threshold for defining the start date of the PCIs and reported it as a variable in the dataset.*

Introduction and elsewhere – literature cited. I suggest that the authors carefully review their citations, as I noted several instances when inappropriate references (e.g., marine, satellite) were used to describe or infer in-lake processes.

*Agreed. We have replaced marine and satellite references with ones relevant to lake processes or, where appropriate, we added an acknowledgement of the source context. For example, see lines 102-103: "… these studies analyzed chlorophyll concentrations derived from satellite observations rather than measured in situ."*

85-92. The presumption that bottom-up controls predominate in spring, whereas top down are most important in summer is not valid. As noted above (and in many many other papers),

grazing control of phytoplankton is strongest during the CWP, and diminishes in summer when slow growing colonial cyanobacteria and algae escape grazer control. (see Carpenter and KItchell Trophic Cascade book etc). Without directly measures of grazing (or actual limitation by physico-chemical processes), the authors are setting up an unnecessary (and likely incorrect) framework. I think they might be better served by using the CWP research as a rationale for expecting 1 or 2 phases of phytoplankton biomass change ('growth windows') as they are well studied and explain the patterns seen in the analysis of Chl. The Plankton Ecology Group (PEG) model may be useful.

*We agree with this comment and now refer to general concepts in lake productivity into the introduction and refrain from setting up a specific interpretative framework. These concepts help provide a broad context to potential users of the database. Note, however, that we now avoid unnecessary speculative (and controversial) interpretations of the patterns and trends in the data. As such, our revised manuscript is foremost a data paper on PCIs and RCIs that can be used by the research community.*

Data and methods. Perhaps I missed it, but I think the paper needs a table which explicitly lays out the data sources without requiring the readers to go to the actual database. As part of this, there should be a good array of summary data (dates samples, resolution, parameters, lakes, etc). While I recognized most of the sources of data, I did not find direct attribution of the data to individual investigators within the paper, which I think is inappropriate.

*We fully agree and have added a summary table and data author citations found in the published metadata to the supplementary materials and references section of the manuscript.*

As part of this, I think the authors need to justify the re-publication of already open- source data. I suspect the rationale is that there is a new analysis of Chl a dynamics, but that in itself creates the problem that the method of Chl analysis has not be validated previously (mainly section 2.2). The simplest solution is to hold back publication of this data paper until the method is reviewed and evaluated. Alternately, the authors need to demonstrate more robustly that the method they use is reasonably artifact free (e.g., the statement that growth periods range 2 to 260 days is highly suspect – see below) and very likely dependent on time series resolution.

*We are presenting a dataset that includes data that has been reformatted and augmented with the estimations of RCI, PCI, and NRCC values. In doing so, we follow common practice to summarize data and calculate additional metrics derived from those data. Our derived dataset is in compliance with the licensing for every original source used.*

[Figure]

*We agree that the range of the PCI length may be influenced by variable sampling resolution. To help users of the data navigate this, we now include the mean time between sampling for the*

*data series in our dataset. Thus, the user will be able to filter the database for a desired temporal resolution, for example by selecting only lakes that are sampled monthly, which is a common sampling frequency for many monitoring programs. Furthermore, we have added the extra figure shown above in the supplementary material. The figure shows that the sampling frequency does not have a strong influence on the PCI length distribution. (Note: the figure compares the distribution of PCI lengths for a) the full dataset, b) the dataset filtered for a maximum of 31 days between samples, and c) the dataset filtered for a maximum of 14 days between samples. While the frequency distribution does change somewhat, the range of PCI length is maintained even when selecting for the data with more frequent sampling.)*

Data compilation – I think the authors need to provide much more information on how the data were harmonized. I know from research in the US LAGOs program (Soranno et al), this is a massive part of the compilation process, taking years in the case of large datasets. Just saying that chemistry was measured (l. 125-130) seems insufficient to me. How did techniques differ, does that affect the findings (or not), are old procedures replaced by new ones, etc.

*We expanded the methods section to provide more details on the harmonization of the dataset; see Section 2.1.1. We refer readers to the augmented source summary table, so that they can easily access the metadata related to the techniques used to generate the source data.*

Similarly, lake selection needs to be better justified.  First, it makes absolutely no sense to include two low latitude lakes, famous or not, in a high0latitude study (remove them; delete l. 130-135).  Second, the authors do not appear to recognize that climate systems (Gulf Steam, NAO, etc.) affect latitudinal gradients, producing much different local conditions in the UK/Scandinavia/EU than in North America (mainly continental Canada).  I understand this is not a full analytical paper, however, the use of latitude is uncritical acceptance and can confuse patterns presented in this paper.

*We have removed these two lakes from the dataset.*

*We agree that latitude alone does not determine local climate conditions. Therefore, we now also identify the actual climate zone of each lake in the revised dataset. However, we still provide the lakes' latitudes in our dataset because they affect the photoperiod.*

Fig. 2 – I like the workflow but is it also possible to see how the number of sites/data density changes through the process?  (how much data is lost at each step). Or that could be in an appendix figure.

*We have updated figure 2 to incorporate the changing data density throughout the workflow.*

141-150. Please provide more information on data manipulation. This is too vague/unclear to allow replication.

*We have reformatted and expanded the methods section to provide more information on the data manipulation process; in particular, Sections 2.1.1 and 2.2. Note that most of the added detail is placed in the supplementary information, which we now refer the reader to within the text (lines 193 and 215-216). Please, also note all the code used to clean, format, and generate the PCI dataset can be freely accessed in a public GitHub repository and on Zenodo.*

153-155. I think the authors need to provide more summary data on the time series themselves. A statistical overview of the resolution seems particularly important, as the vast majority of sites would be sampled at weekly or longer intervals, so would be expected to have substantial restrictions on the detection of growth window onset and duration. Monthly resolution could, arguably, make the growth window meaningless.

*As suggested by the reviewer, we have expanded the data source summary table to include information about the sampling resolution of each data series. We have also expanded the information about the derived data (i.e., RCI, PCI, NRCC) by including a column containing the sampling resolution of the corresponding source data. We have included a summary figure (Fig 4b) in the manuscript illustrating the distribution of sampling resolution within the compiled data sources to be transparent about the variability in source data sampling resolution.*

SSR. The use of these data need much better justification. First, as shown from the map in Fig. 1 – the vast majority of lakes are not located very close to SSR stations. Second, cloud cover would be expected to greatly influence the receipt of solar irradiance, but would not be recorded well for individual lakes. I *like* the attempt to use SSR data to explain variation in growth windows, but find the approach was largely unsuccessful – likely because the data were inappropriate. I think the authors need to better justify the use of SSR or drop the analysis.

*We agree with the reviewer that the distances between the lakes and the SSR stations is a major limitation. However, SSR is an essential variable when considering trends in lake productivity. Our dataset on SSR is the result of our best efforts to collate the data that are available. By providing the distance between a lake and the corresponding SSR station, the user can make up their own mind as to whether meaningful inferences can be made or not. The user can also use a cut-off distance to filter the dataset. We hope that our dataset will call attention to the need for more direct over-lake SSR measurements, particularly in view of the mounting evidence that secular changes in SSR are modulating photosynthetic rates in terrestrial ecosystems. In the revised manuscript we have added a figure showing the frequency distribution of lake-SSR station distances, and state that "Users are therefore advised to consider this limitation [of distance] when making use of the SSR values in our dataset" (lines 314-315). We also added a sentence in the text to highlight the potential impact of cloud cover and aerosol variability on the representativeness of the SSR measurements: "...in a significant number of cases, the actual mean SSR during a PCI may differ from the in situ mean SSR reported here, due to differences in cloud cover and levels of atmospheric aerosols (among other factors)." (lines 311-313).*

2.1.3 – the summary of lake characteristics is pretty limited. This would seem to be important if the data are to be used in other analyses (as inferred).

*Information on the lake characteristics can be found in the summary metadata files accompanying our dataset. We have added a clarifying statement on line 183 directing readers towards these summary files so they can access this information: "The reader is referred to the "lake summary" file in the supplementary information for details on the lake characteristics."*

Section 2.2. Detection growth windows. The paper hinges on readers accepting 0.05 ug Chl /L/d as a critical threshold for signifying 'growth'. Its use because it was the median value in lakes

with a CWP (productive; 2 growth phases) is one possible justification; however, the use of the same threshold in oligotrophic lakes seems inappropriate as Chl values may be up to two orders of magnitude lower than in eutrophic systems. Basically, use of a single static metric over all lake types seems unjustified.

One possibility would be to do a sensitivity analysis. The selection of a single threshold immediately raises the question of how the data patterns would change in a different threshold were used. Normally (in a methods paper), the authors would conduct some form of sensitivity analysis to demonstrate that the findings were (not) robust to the precise value used in the study. (but again, the authors need to decide whether this Is a data paper or a full analytical report).

*We appreciate the thorough comments and suggestions for selecting a threshold value for the start of the growth window (which we have now renamed as "period of chlorophyll-a increase", PCI). To address this concern, we have first replaced the absolute rate of chlorophyll-a increase RCI as the threshold metric with the normalized rate of change in chlorophyll-a concentration (NRCC). We highlight in lines 263-276 that, in contrast to RCI, NRCC is a relative rate, which can be compared between lakes across a range of standing stock of chlorophyll-a and trophic classes.*

*Second, we have now included in our supplementary material the results of our sensitivity analysis of the derived PCI and RCI distributions to the imposed threshold NRCC value that is used to define the start of the PCI. A full explanation of the calculation of the normalized rate and the sensitivity analysis is included as online supplementary material and referred to in the text on lines 190-192. Briefly, we have performed a sensitivity analysis for NRCC threshold values ranging from 0 to 1.2 $d^{-1}$. Results from a Kruskal-Walis test showed that the PCI start day and PCI length change significantly when the threshold NRCI approaches zero. Based on the results of the sensitivity analysis, we have selected a threshold value of 0.4 $d^{-1}$. This relatively high value reduces the risk of erroneously including the lead-up time to the PCI. Hence, the calculated rate RCI is more likely to be a measure of the exponential increase in algal biomass.*

Data set. As noted about, some summary characteristics of the lakes (morphometry), Chl time series (sampling resolution, etc) and chemistry would be useful to interpret the patterns seen in l 215-224.

*As stated earlier, we direct the readers on lines 179-180 to the lake summary files in the supplementary information where they can find this information and related data (including the sampling resolution): "The reader is referred to the "lake summary" file in the supplementary information for details on the lake characteristics."*

220-221 and elsewhere. Try to be less declarative about the reason for unmeasured patterns. In North America, latitude is correlated inversely with human populations and activity. To an extent, this is true in the European continent.

*Agreed. We have removed the sentence and reworded other declarative statements throughout the text.*

L 229-231 again cause of patterns two speculative (and see comments about CWP above)

*Agreed. We have removed the speculative statement.*

253-263. This paragraph is overly speculative, as it is based on marine gradients, not those of lakes. See work of John Smol for patterns in Northern Canada.

*Agreed. We replaced references to marine gradients with the suggested references to the work of John Smol.*

282-285. Again, this is over-extrapolated (from a fossil study that does not prove mechanism). More importantly, the analysis of SSR is vague or unreproducible. "Sensitivity" to SSR is never defined, and there is no obvious relationship in panels of Fig. 9. This whole section in unconvincing because there is no clear relationship among variables, and because the SSR sites are not particularly close to the study lakes.

*Agreed. We removed the reference to the McGlue et al study from the paper. We have also replaced the interpretative SSR correlation analysis with Section 4.3, a description of trends seen in the SSR data within our dataset.*

Key findings. In my opinion, too many of the findings infer mechanisms which were not proved rigorously in the paper. It's fine to review the main patterns, but it's inappropriate to infer causal mechanisms from the sorts of analyses presented. Again, I think the authors are trying to move this from a data paper to a methods or interpretive paper. While this may make the paper more scientifically interesting, I don't find it appropriate for the format of a data paper.

*We have removed this section from the text in favour of focusing on presenting the trends as examples of observations that can be extracted from the dataset (without going into speculative interpretations).*

---

## Author Comment (AC2)

**Comment on essd-2021-329**
**Response to Referee #2**

Referee comment on "Chlorophyll-*a* growth rates and related environmental variables in global temperate and cold-temperate lakes" by Hannah Adams et al., Earth Syst. Sci. Data Discuss., https://doi.org/10.5194/essd-2021-329-RC2, 2022

This manuscript describes a valuable data set on chlorophyll a dynamics and associated environmental variables for lakes covering a wide range of limnological conditions and localities (mostly north temperate). Specifically, the authors identified periods of increasing chlorophyll *a* concentrations, and they calculated the increment rates for such periods in each lake. This approach and the compilation of calculated rates has resulted in a derived dataset that will be of interest to many researchers.

*We thank the reviewer and appreciate their suggestions and comments for improving our manuscript; all their suggestions have been considered in the revision of the manuscript. Below, we provide the answers to the comments and questions raised by the reviewer and all the modifications that have been incorporated in the revised version of the article.*

The description of this dataset would likely be appropriate to ESSD, but I have two concerns:

1. Although 'growth rates' appears in the title and manuscript, the estimates produced are not growth rates *per se*, but are net increment rates. In the phytoplankton literature and the ecological literature in general, the term growth rate refers to the parameter r in the exponential growth of populations: dN/dt = rN. For phytoplankton, N = cell number or a proxy (such as chlorophyll), and r is the specific growth rate with units of inverse time ($d^{-1}$). In this study, I don't think that the authors fitted curves to the data (except for smoothing), but instead took chlorophyll values at the start and end of each growth window, subtracted the values, and divided by the time interval (this my interpretation; the exact calculation method needs to be more explicitly stated in the manuscript). The resultant parameter is therefore in linear increment units ($\mu g$ Chl $a$ $L^{-1}$ $d^{-1}$) not growth rate units ($d^{-1}$), and it is misleading to call this a phytoplankton growth rate. In fact, sections of illustrative Fig. 3 do look more like exponential growth rather than linear increments, notably March-April and July-August.

Additionally, this calculated rate measures not only phytoplankton growth but also losses, and it is therefore a **net** rate of increase. This may be why the changes may be near- linear (e.g., August-September) rather than exponential (also, these are averages, with phytoplankton having different net growth rates down though the mixed layer, different parts of the lake, etc.).

*We agree that the net rate of chlorophyll-a increase includes both growth and loss terms. We also agree that the terminology used in our original manuscript may result in unnecessary confusion. To avoid ambiguity in terminology, we no longer use the term "growth" when referring to the durations and rates of chlorophyll-a increase. Specifically, we have renamed:*

- *"Growth rate" as "rate of chlorophyll-a increase" (RCI), defined on lines 115-116*
- *"Growth window" as "period of chlorophyll-a increase" (PCI), defined on line 107*
- *"Specific growth rate" as "normalized rate of change in chlorophyll-a concentration" (NRCC), defined on lines 182-183*

*Renaming the "specific growth rate" to "normalized rate of change in chlorophyll-a concentration" (NRCC) avoids the assumption that the rate of increase in chlorophyll-a*

*concentration must be strictly proportional to the algal biomass concentration.*

*We have reworded our explanation about what the calculated rate of chlorophyll-a increase (RCI) represents and what the limitations of this metric are in lines 263-276. We have also highlighted in the text that the normalized rate, NRCC, is a relative rate which facilitates the comparison between lakes of different trophic status, whereas the RCI will vary systematically between lakes with different standing stocks of chlorophyll-a (lines 263-276). We have calculated the NRCC to use as the threshold for defining the start date of the PCIs and reported it as a variable in the dataset.*

*We now explicitly state on line 115 that RCI is a net rate, and we have also added a sentence in the introduction explaining that the net chlorophyll concentrations observed (and therefore also the net rates) are controlled by multiple process (lines 80-84):*
*"Intra-annual fluctuations in lake chlorophyll-a concentration result from the interactions of multiple variables and processes including grazing by zooplankton, competition between algal species with different growth strategies and chlorophyll-a contents, and changes in temperature, light, and nutrient availability."*

*To provide a more stepwise explanation of the calculation method for the PCI, RCI and NRCC metrics, we have expanded our explanation in the text in Section 2.2 and now refer readers to the relevant supplementary information.*

Without making this distinction between net rates of phytoplankton change (as estimated) versus phytoplankton growth rates (not estimated), it is easy to be led astray in interpreting the data. For example, the authors find that the net increment rate is lower in higher latitude waters:

Line 331: "Chlorophyll-a growth rates increase with nutrient availability while they decrease at higher latitudes due to cooler temperatures and lower SSR."

But higher latitude waters are largely ultra-oligotrophic. This means that phytoplankton increment rates in absolute terms ($\mu$g Chl-a $L^{-1}$ $d^{-1}$) can never be large; there is not enough standing stock (nor available nutrients) to allow a large absolute increment, as opposed to a southern eutrophic lake where even a 5% increase would be huge in absolute terms (this also biases the analyses towards eutrophic waters, with the cutoff expressed in absolute rather than relative terms; line 180).

On the other hand, the specific growth rate of high latitude, cold-adapted phytoplankton could be rapid (as in algal blooms in the polar oceans) with the growth supported by nutrient recycling processes, and population size kept in check by grazing and other loss processes, as well as capped by TP and other nutrients.

I do think that the estimates and window approach are very interesting, as are the trends, but the terminology needs to be re-thought. The flip side of the question is also interesting, the net rates of chlorophyll decrease. This same approach (but for periods of decreasing chlorophyll) could be used to identify periods of sedimentation (storage fluxes) and/or high grazing intensity. The paper could be retitled "Net rates of chlorophyll-*a* change and…' with the abstract explaining that these are net rates of linear increase or decrease. Or the application to net loss rates could be just mentioned in the Discussion, without the need to update the database.

*We followed the reviewer's advice and changed the terminology to be more accurate. As mentioned in our response to the reviewer's previous comment, we have renamed the calculated metrics to remove the word "growth". We have also added clear definitions of the rate*

*terminology we use in the text indicating that they are "net" rates of increase of the chlorophyll-a concentration (absolute and normalized) on lines 115 and 182-183. Furthermore, lines 263-276 now outline the potential for inter-lake comparisons using the absolute rates (RCI). These lines provide an explanation as to the differences between RCI and NRCC and potential biases with the use of RCI.*

*We have evaluated the sensitivity of the derived values in the dataset (e.g., RCI, PCI start and end dates) to the threshold rate value that we used to define the start of the PCI. The full explanations of the calculation of the normalized rate (NRCC) and the sensitivity analysis are now included as online supplementary material. Using NRCC as the threshold means that the comparison of rates between lakes accounts for differences in phytoplankton standing stock between lakes of different trophic status. We refer the reader to this supplementary material in line 192 of the revised manuscript.*

*We agree with the reviewer that our rate approach would be interesting to apply to periods of chlorophyll-a decrease, in addition to periods of increase. While this avenue of inquiry would certainly be interesting, we feel it is outside of the scope of the current manuscript, which above all is intended to be a data description paper.*

2. Several sections of this manuscript read more like a scientific research article than a data description paper. For ESSD, it seems like it would be better to focus on the rationale (the current Introduction, which reads very well), the methods, and the resultant dataset, and leave the questions, hypotheses, trend analyses and interpretations to a paper for publication in a limnological journal that then refers to this article for the dataset methodology and to Adams et al. (2021) for the complete compiled data (which I verified to be available for download and well organized; I see the data sources are given in the readme file but it would be useful to have the citations for the original limnological data as the final column in the lake_summary data file).

This question of how much interpretation to include would be best to discuss with the ESSD editors. In checking the website, I see that some recent articles go beyond a description of the data to include trend and spatial analyses, for example:

https://essd.copernicus.org/articles/14/517/2022/

https://essd.copernicus.org/articles/14/463/2022/

while others are more exclusively focused on describing a dataset, for example:

https://essd.copernicus.org/articles/14/449/2022/

*We agree that maintaining a balance between data description and analysis in an ESSD article is challenging. We thank the reviewer for pointing us toward ESSD articles that go beyond just describing their data. These articles were helpful during the revision of our manuscript. In the revised manuscript we have reduced the causal interpretations of the trends shown. Instead, our focus is on illustrating the type of trends that can be extracted from the dataset. Thus, we have removed much of the interpretative text related to the figures, instead highlighting the features of the trends and relationships.*

---

## Author Response (AR2)

**Comments to the author**:

Dear Authors

Thank you for your replies to the reviewers, for the throughout revision of your manuscript and dataset and sorry for the delay in our editorial feedback. We are pleased to tell you that your publication is accepted for publication in ESSD with some minor revisions.

*Thank you for your careful review of the manuscript, we agree with all your suggestions and have edited the manuscript accordingly.*

The referees asked for overview on data sources and the variables in the lake summary. In your manuscript text it is indicated that this information is part of the supplementary datasets (it is not a supplementary information attached to the ESSD manuscript because this is all metadata linked to the data and an integral part of the dataset publication).

*We agree that the locations of the data sources are not clear, this information has been added to the manuscript as suggested below:*

Your data publication is very well organised and datasets are userfriendly, the dataset history and data sources and licences are well documented.

Here. readers of the ESSD manuscript would need partly more information on the data sources and data content provided in the manuscript text without that they would need to organise it separately via data download. We readers will appreciate to have this information as overview available in the manuscript:

i) a table with a list of sources for Chl-a in the selected lakes, e.g. it exists already in the README in table format. You could cite it in chapter 2.1.1 and add it in the ESSD manuscript appendix

*We have added a table summarizing the sources for the lake data (Table A1), and referred the readers to the table in chapter 2.1.1 (lines 131-132 and 137)*

ii) a table with a list of sources for SSR, you could cite it in chapter 2.1.2

*We have added a table with this information to the appendix (Table A2) and cited it in the main text (lines 137 and 164).*

iii) a table overview of the variables of the associated lake data (e.g., lake depth, surface area, volume, climate zone) with data sources, also as table in the appendix. you could cite it in chapter 2.1.3,
you can also indicate that these metadata collection is published as part of your data publication (Adams et al., 2021) instead of the sentence in the manuscript 'The reader is referred to the "lake summary" file in the supplementary information for details on the lake characteristics.' that sounds misleading

*We replaced the sentence with a new one indicating that the metadata can be found along with the published data (lines 180-181). We have also added a table summarizing the associated lake data to the appendix (Table A4) and cited it in chapter 2.1.3 (lines 181-182). We also included a table summarizing the databases we used as sources for these data in the appendix (Table A3).*

chapter 2 data and methods: lake name - in the dataset publication you describe it as reformatted from original file - how did you standardise, / agreed on the lake name? could you add there a sentence in the text.

*We added a sentence to the text, clarifying how the lake names were standardized (lines 157-158).*

Many thanks and we look forward to your final manuscript,
Best wishes, Birgit Heim